

# On the importance of spatial scales on beta diversity of coral assemblages: a study from Venezuelan coral reefs

Emy Miyazawa[1], Luis M. Montilla[1,2], Esteban Alejandro Agudo-Adriani[1,3], Alfredo Ascanio[1,4], Gloria Mariño-Briceño[1] and Aldo Croquer[1,5]

[1] Laboratorio de Ecología Experimental, Universidad Simón Bolivar, Caracas, Venezuela
[2] Integrative Marine Ecology Department, Stazione Zoologica Anton Dohrn, Naples, Italy
[3] Department of Biology, University of North Carolina at Chapel Hill, North Carolina, United States of America
[4] Department of Biology, Miami University of Ohio, Oxford, OH, United States of America
[5] Centro de Innovación Marina, The Nature Conservancy, Punta Cana, Dominican Republic

## ABSTRACT

Estimating variability across spatial scales has been a major issue in ecology because the description of patterns in space is extremely valuable to propose specific hypotheses to unveil key processes behind these patterns. This paper aims to estimate the variability of the coral assemblage structure at different spatial scales in order to determine which scales explain the largest variability on β-diversity. For this, a fully-nested design including a series of hierarchical-random factors encompassing three spatial scales: (1) regions, (2) localities and (3) reefs sites across the Venezuelan territory. The variability among spatial scales was tested with a permutation-based analysis of variance (Permanova) based on Bray-Curtis index. Dispersion in species presence/absence across scales (i.e., β-diversity) was tested with a PermDisp analysis based on Jaccard's index. We found the highest variability in the coral assemblage structure between sites within localities (Pseudo-$F$ = 5.34; $p$-value = 0.001, CV = 35.10%). We also found that longitude (Canonical corr = 0.867, $p$ = 0.001) is a better predictor of the coral assemblage structure in Venezuela, than latitude (Canonical corr = 0.552, $p$ = 0.021). Largest changes in β-diversity of corals occurred within sites ($F$ = 2.764, df1= 35, df2 = 107, $p$ = 0.045) and within localities ($F$ = 4.438, df1= 6, df2 = 29, $p$ = 0.026). Our results suggest that processes operating at spatial scales of hundreds of meters and hundreds of kilometers might both be critical to shape coral assemblage structure in Venezuela, whereas smaller scales (i.e., hundreds of meters) showed to be highly-important for the species turnover component of β-diversity. This result highlights the importance of creating scale-adapted management actions in Venezuela and likely across the Caribbean region.

## INTRODUCTION

The importance of scales in ecology has been largely acknowledged for decades (*Schneider, 2001*; *Mac Nally & Quinn, 1998*; *Wiens, 1989*). *MacArthur (1972)* and *Levin (1992)* assays

Corresponding authors
Emy Miyazawa, 12-11222@usb.ve
Aldo Croquer, acroquer@usb.ve

deeply influenced modern ecologists by promoting the view that ecological processes act at a variety of spatial and temporal scales, and they generate patterns that may differ from those at which processes act (*Chave, 2013*). Today it is known that ecological dynamics tend to be stochastic at small scales, but variability is conditional on the resolution of description (*Chave, 2013*; *Levin, 1992*). Furthermore, there has been an increased recognition that the problem of scale at which ecological processes act, should be considered as critical if it is wanted to produce general predictions about patterns in space and time (*Chave, 2013*). Thus, modern ecological thinking agrees that in order to understand a system (e.g., a community), it is important to study it at the appropriate scale (*Chave, 2013*).

It is clear that increasing consideration of scale is helping to address a key issue in ecology: the question of what influences the distribution and abundance of organisms (*McGill, 2010*). Species distributions depend on four important processes: (1) climate, (2) species interactions, (3) habitat structure and (4) dispersal capabilities, each one operating with different strength at a range of spatial scales (*McGill, 2010*). Generally, the presence or absence of organisms within a community may depend on rare or large-scale (region-specific) dispersal and colonization events, while local abundance is more a function of frequent, fine-spatial scale processes such as biotic interactions and habitat heterogeneity, e.g., *Ricklefs (1987)*. This implies that communities are structured by both abiotic and biotic factors nested along different spatial scales which often occur along environmental gradients (*Johnson & Goedkoop, 2002*; *Whittaker & Heegaard, 2003*). Concomitantly, the species richness of a community is also expected to be highly dependent on spatial scales evaluated (*Barton et al., 2013*; *Field et al., 2009*; *Melchior, Rossa-Feres & da Silva, 2017*; *Whittaker, Willis & Field, 2001*).

Coral reefs are one of the most complex and diverse ecosystems of the planet. Reef species diversity has been estimated from 600,000 to more than 9 million species worldwide (*Plaisance et al., 2011*; *Reaka-Kudla, 1997*). The habitat and shelter for the majority of these species is largely provided by scleractinian corals (*Alvarez-filip et al., 2009*). There is compiling evidence indicating that ecological processes controlling the structure of coral assemblages (e.g., substrate availability, recruitment, competition, and herbivory) are strongly dependent on spatial scales (*Pandolfi, 2002*). In addition, oceanographic processes which partly define the environmental setting of a reef are also extremely variable within habitats, across sites, reef systems, and regions (*Chollett, Mumby & Cortés, 2010*; *Eidens et al., 2015*). Furthermore, biological and environmental factors may interact with each other to produce different patterns in species distribution across several spatial scales. In consequence, understanding the underlying factors controlling the coral species richness in a reef is not a simple task (*Edmunds, 2013*; *Eidens et al., 2015*; *Karlson & Cornell, 1999*) for it is a multi-scale problem (*Komyakova, Jones & Munday, 2018*).

Total species richness of a region, frequently named gamma diversity ($\gamma$), can be partitioned in two components: (1) $\alpha$-diversity (i.e., the number of species by site), and (2) $\beta$-diversity (i.e., the variation in the species identities from site to site, *sensu Whittaker (1960)* and *Whittaker (1972)*. For decades, ecologist have debated ways to estimate and interpret $\alpha$ and $\beta$-diversity; but in recent years, the study of $\beta$-diversity has gained a lot

of interest for it is what actually makes assemblages of species more or less similar to one another at different places and times (*Anderson et al., 2011*; *Vellend, 2010*). Many different measures of $\beta$-diversity have been introduced, but there is no overall consensus about which ones are most appropriate for addressing particular ecological questions (*Jurasinski, Retzer & Beierkuhnlein, 2009*; *Tuomisto, 2010a*; *Tuomisto, 2010b*). *Anderson et al. (2011)* distinguished two types of $\beta$-diversity: (a) turn-over and (b) variation. Turn-over refers to changes in community structure among sampling units distributed along well-defined environmental gradients, whereas variation portrays variability in species composition among sample units within a given spatial or temporal extent, or within a given category of a factor (such as a habitat type or experimental treatment). On the other hand, *Baselga (2010)* partitioned the total $\beta$-diversity into two components: (1) nestedness, i.e., when the species composition of sample units with low richness represent a subset of the species found in the richest sample units, and (2) species replacement, i.e., a turn-over of species (*Gaston et al., 2007*; *Leibold et al., 2004*; *Svenning, Fløjgaard & Baselga, 2011*). Regardless the point of view, the study of each of these components is relevant to understand processes that control ecological communities and a range of ecosystem functions (*Arias-González, Legendre & Rodríguez-zaragoza, 2008*; *Harborne et al., 2006*).

While spatial patterns of $\gamma$ and $\alpha$-diversity of coral assemblages have been studied extensively; only few studies have focused on measuring $\beta$-diversity, (*Arias-González, Legendre & Rodríguez-zaragoza, 2008*; *Connell et al., 2004*; *Harborne et al., 2006*). This is the case of Venezuela, where most of the papers published to date have only been focused on site descriptions based on species composition and abundance, whereas the influence of spatial variation across different scales on coral assemblages remains poorly explored. The Venezuelan coast is highly heterogeneous with clear longitudinal environmental gradients (*Chollett, Mumby & Cortés, 2010*; *Miloslavich et al., 2003*) which are deeply influenced by up-welling regimes that play an important role for the distribution of marine biodiversity (*Miloslavich et al., 2010*). In fact, algal communities in rocky shores (*Cruz-Motta, 2007*) and sessile organisms associated to mangrove roots have been found to vary at different spatial scales along the Venezuelan coast (*Guerra-Castro, Cruz-Motta & Conde, 2011*). Thus, it should not be surprising to find coral assemblages to be extremely variable across spatial scales in Venezuela. We expected that greater changes in community structure and $\beta$-diversity of coral assemblages will occur at scales of thousand of kilometers (i.e., between the eastern and western regions) and within sites (i.e., hundreds of meters). This is because of existing contrasting environmental settings driven by upwelling spots that have been described along the Venezuelan coast line (*Chollett, Mumby & Cortés, 2010*; *Miloslavich et al., 2003*).The goal of this study was two-fold: (1) to quantify spatial variation of coral assemblages from hundreds meters to hundreds of kilometers, and (2) to determine if there are patterns of $\beta$-diversity across these scales.

## METHODS

### Study area

We conducted a multi-scale sampling design comprising coastal areas as well as continental and oceanic islands (Fig. 1). Specifically, seven localities were sampled along the Venezuelan
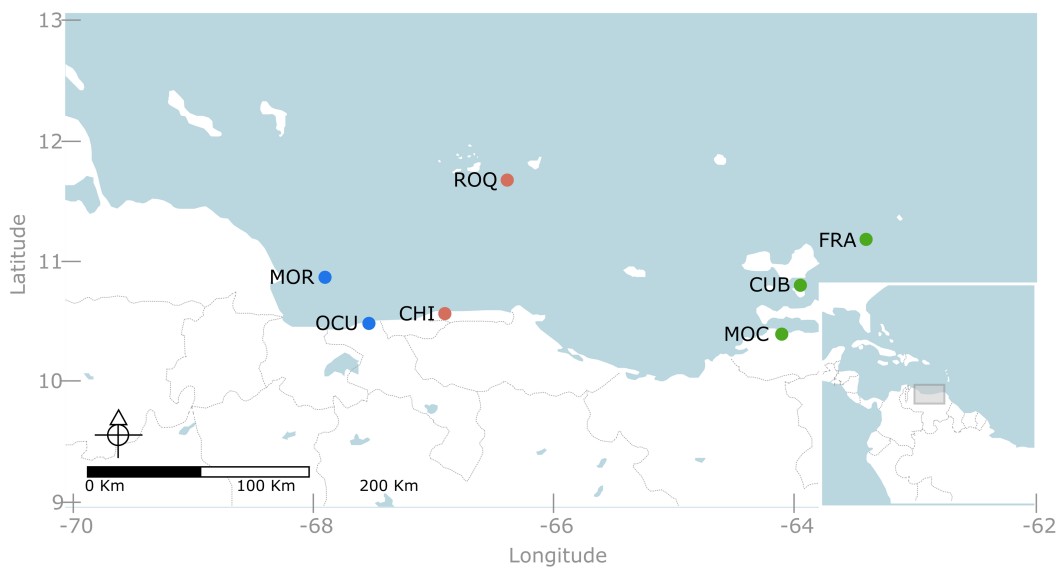

**Figure 1** **Map of the Venezuelan coast with the seven locations used in this study.** Western region, represented in blue, include: MOR = Morrocoy National Park and OCU = Ocumare de la Costa. Central region, represented in red, include: ROQ = Archipielago Los Roques National Park and CHI = Chichiriviche de la Costa. Eastern region, represented in green, included MOC= Mochima National Park, CUB = Cubagua and FRA = Los Frailes.

territory encompassing three contrasting regions (Fig. 1). The western region, included two localities: (1) Morrocoy National Park (MNP) and (2) Ocumare de la Costa. The former is a continental reef system formed by a group of keys and lagoons surrounded by fringing and patch reefs located nearby mangroves and seagrass beds (*Weil, 2003*); whereas the latter is a small bay protected by reef barriers with seagrass and mangroves dominating the inner and shallower habitats. Likewise, the central region, entailed two localities: (3) Los Roques National Park (LRNP) which is an oceanic archipelago with a central lagoon, characterized by extensive reef banks/patches and two large coralline barriers located south and east of the archipelago (*Weil, 2003*); and (4) Chichirivivhe de la Costa, a location of rocky reefs with steep slopes and scattered coral assemblages. Finally, in the eastern region three localities were included: (5) Mochima National Park (MoNP), (6) Los Frailes and (7) Cubagua. In Mochima, seagrass beds and mangroves border a rocky coastline with steep slopes and fringing reef communities. Los Frailes and Cubagua are islands lying at the continental shelf and dominated by small patch reefs with scattered coral assemblages bordering their coastlines. The whole eastern coast of Venezuela and its continental islands are subjected to seasonal upwelling due to its connection with the Cariaco trench (*Weil, 2003*). The selection of these locations aimed to cover the vast majority of reef habitats described for Venezuela (*Weil, 2003*). Permits for taking pictures at marine protected areas was given by Ministerio del Poder Popular para el Ecosocialismo y Aguas - Dirección General de Diversidad Biológica under the office no. 0033.

## Experimental design

A fully-nested design encompassing three hierarchical-random factors (i.e., site, locality, and region) was used to determine spatial variation on coral assemblage structure (i.e., absolute cover of coral species) and $\beta$-diversity from hundreds of meters (sites) to hundreds of kilometers (region). The factor region encompassed three levels (West, Center, and East); nested within region there were two/three localities, four to seven reef sites within each locality, and four 30m-long transects within each site, understood as the operational unit.

## Benthic surveys

At each reef site, benthic surveys were conducted during 2017 and 2018, following the guidelines outlined by the Global Coral Reef Monitoring Network-Caribbean (GCRMN) (*GCRMN, 2014*) with slight modifications. In order to increase the number of sampled sites, we surveyed four instead of five 30 m-long transects parallel to the shoreline following the bottom contour between 8–10 m depth. Transects were set randomly, with the first transect being always layout at the first spot of diving. From that point, each transect was moved up or down from the first transect. Distance among transects varied from 5 to 6 m, so each operational unit was inter-spaced across the sampled reef habitat. For each transect, 15,80 × 90-cm photos were taken every other meter to determine the benthic community structure ($N = 60$ photos per site). A reference frame was used in the field to calibrate each photograph in the laboratory for further analysis of benthic cover.

## Analysis of photo-quadrats

The photo quadrat analysis was performed using PhotoQuad (*Trygonis & Sini, 2012*). For this, every coral was identified to species level and the percentage cover was estimated from 25 points randomly set in an area of approximately 7,200 cm$^2$. From the analysis of photo quadrats, we obtained two matrices: (1) absolute cover of coral species and (2) coral species presence/absence. Data cleaning and quality control were performed using R (*R Core Team, 2019*). Thus, coral cover estimates were done from a randomly-selected sample composed of 375 points per transect (15 photos × 25 points = 375).

## Data analysis

A $Log(x + 1)$ transformation was applied to the data and the Bray-Curtis index was used to build the similarity matrices. Hypothesis test were performed with a permutation-based analysis of variance (PERMANOVA). A test of homogeneity of dispersions (PermDisp) was performed before running the PERMANOVA to verify dispersion of data across different spatial scales. For the majority of the spatial scales the PermDisp confirmed homogeneity in dispersion except for sites (Supplemental Information 2). However the PERMANOVA test has been shown to be robust to identify dispersion versus location effects for balanced designs like the one presented here (*Anderson & Walsh, 2013*).

With the PERMANOVA, we tested the spatial variability of coral assemblage structure (i.e., absolute abundance of each species) from meters to hundreds of kilometers. The variance components were estimated depending on each source of variation in the analysis following the procedures outlined by *Clarke & Gorley (2006)*; *Anderson, Gorley & Clarke (2008)*. Spatial patterns (if any) were visualized by using an Analysis of Principal

Coordinates (PCO) aimed to represent the position of centroids of each site in a Bray-Curtis space. Also, a Similarity Percentage Analysis (SIMPER) was carried out at each spatial scale to identify and estimate the species that contribute the most to the patterns observed. SIMPER results were represented as heat maps to show the relative contribution of each coral specie to the average Bray-Curtis similarity. Additionally, a Canonical Analysis of Principal Coordinates (CAP) based on Bray-Curtis similarity index was carried out. We used the scale that contributed the most to the variation in the dissimilarity of coral species to determine the correlation between the coverage of scleractinian corals and their latitudinal and longitudinal position. For coral cover we used the same variables as in the PERMANOVA, and the latitude and longitude were understood as proxies of distance respect to the coast and position along the coast, respectively. All analyses were carried out using Primer 6 - Permanova + (*Clarke & Gorley, 2006*; *Anderson, Gorley & Clarke, 2008*).

Differences on $\beta$-diversity were tested across different spatial scales using a test of homogeneity of dispersions (PermDisp) *Anderson, 2006*) based on presence/absence Jaccard index. To keep the nested design we used transects to test dispersion across sites, the composition of sites to test dispersion across localities, and the composition of localities to test dispersion across regions. The Jaccard index was then split into the components of nestedness and turn-over using the *BetaPart* package in R (*Baselga & Orme, 2012*).

Due to the life history strategies of scleractinian corals (i.e., large colonies of a single species may often occupy an entire quadrat), rare species in the assemblages might have been underestimated (*Chao et al., 2014*). To verify this, we estimated sampling coverage (i.e., the relationship of interpolation and extrapolation of the effective number of species in the community) (*Hsieh, Ma & Chao, 2016*; *Chao et al., 2014*). This methodological approach allows statistical inferences without the bias of underestimated rare species in assemblies and/or assigning them the same weight as abundant species (*Chao et al., 2014*). The sampling coverage was calculated using the *iNEXT* package in the R software (*Hsieh, Ma & Chao, 2016*). We found that 80% of the sites had a sample coverage greater than 0.7 (Table 1, Supplemental Information 3), which is considered a proper sampling effort to describe species assemblages by *Hsieh, Ma & Chao (2016)*.

## RESULTS

### Patterns of coral assemblage structure

The results show that species composition and abundance of corals in Venezuela varied across different spatial scales. The greatest variability was found at the scale of sites (Pseudo-$F = 5.34$, $p$-value $= 0.001$) which accounted for 35.1% of the total variance (Table 2). The scale of hundreds of kilometers was the second most important source of variation in the analysis (Pseudo-$F = 3.35$, $p$-value $= 0.01$, CV $= 21.35\%$ Table 2). This result indicates that coral assemblages in Venezuela only vary by 21.35% at the scale of region. Also, we found statistical significance at the scale of locations within regions, explaining 11.42% of the total variance (Table 2). Thus, our results indicate that coral assemblages are much variable at small to medium scales (i.e., from hundreds of meters to tens of kilometers) rather than hundreds of kilometers (i.e., regions) alone.

**Table 1 Sampling coverage for each reef site.**

| Locality | Site | Replicates | Incidence | Richness | Sampling coverage |
|---|---|---|---|---|---|
| Morrocoy | Boca Seca | 4 | 12 | 6 | 0.900 |
| | Bajo Caimán | 4 | 9 | 4 | 0.952 |
| | Medio | 4 | 13 | 7 | 0.736 |
| | Mero | 4 | 16 | 10 | 0.796 |
| | Norte | 4 | 20 | 7 | 0.979 |
| | Sombrero | 4 | 24 | 11 | 0.836 |
| | Sur | 4 | 18 | 8 | 0.885 |
| Ocumare | Ciénaga este | 4 | 15 | 8 | 0.706 |
| | Ciénaga interna | 4 | 24 | 11 | 0.779 |
| | Ciénaga oeste | 4 | 25 | 10 | 0.917 |
| | Guabinitas | 4 | 16 | 7 | 0.925 |
| Los Roques | Cayo Agua | 4 | 25 | 11 | 0.893 |
| | Boca de Cote | 4 | 33 | 12 | 0.937 |
| | Dos Mosquises sur | 4 | 21 | 10 | 0.790 |
| | Madrisqui | 4 | 17 | 7 | 0.912 |
| | Pelona de Rabusqui | 4 | 14 | 6 | 0.839 |
| | Salinas | 4 | 23 | 10 | 0.851 |
| | La Venada | 4 | 7 | 4 | 0.679 |
| Chichiriviche de la costa | La Pared | 4 | 21 | 9 | 0.901 |
| | Media Legua | 4 | 15 | 9 | 0.640 |
| | Punta de Media Legua | 4 | 17 | 9 | 0.741 |
| | Petaquire | 3 | 11 | 7 | 0.636 |
| | Playa Tiburón | 4 | 20 | 7 | 0.979 |
| | Punta Mono | 4 | 19 | 8 | 0.828 |
| Mochima | San Agustín | 4 | 1 | 1 | 1.000 |
| | Blanca | 4 | 23 | 9 | 0.893 |
| | Carabela | 4 | 2 | 2 | 0.400 |
| | Punta Cruz | 4 | 14 | 6 | 0.893 |
| | Gabarra | 4 | 11 | 5 | 0.864 |
| | Garrapata | 4 | 20 | 7 | 0.940 |
| Cubagua | Charagato | 4 | 6 | 4 | 0.625 |
| | Punta Conejo | 4 | 6 | 4 | 0.625 |
| | La Muerta | 4 | 6 | 4 | 0.625 |
| Los Frailes | Cominoto | 4 | 6 | 2 | 1.000 |
| | La Pecha | 4 | 1 | 1 | 1.000 |
| | Puerto Real | 4 | 8 | 2 | 1.000 |

With over 70% of the total variance explained by the first three PCO axes, localities within regions ordinate according to changes in cover of coral species (Fig. 2). Our results show that the eastern and central regions are much more similar to each other concerning the western region. The SIMPER analysis showed clear patterns defining regions and locations across Venezuela, however, sites showed variable species composition and cover (Fig. 3).

**Table 2  Three-way permutation-based analysis of variance (PERMANOVA) based on Bray-Curtis Similarity to test differences in coral assemblage structure.**

| Source | df | SS | MS | Pseudo-F | P(perm) | Unique perms | %CV |
|---|---|---|---|---|---|---|---|
| Region | 2 | $7.11 \times 10^4$ | 35529 | 3.35 | 0.01 | 998 | 21.35 |
| Locality | 4 | $4.09 \times 10^4$ | 10219 | 2.28 | 0.002 | 998 | 11.42 |
| Site | 29 | $1.30 \times 10^5$ | 4483 | 5.34 | 0.001 | 994 | 35.10 |
| Residuals | 107 | $8.99 \times 10^4$ | 840.17 | | | | 32.14 |
| Total | 142 | $3.36 \times 10^5$ | | | | | |

Overall, Western region was largely composed of **Orbicella faveolata** reefs, whereas the eastern reefs were dominated by **Pseudodiploria strigosa** (Fig. 3). On the other hand, across the Central regions which included Oceanic and Coastal Reefs, mixed coral communities were found. These species accounted for more than 75% of dissimilarities across localities, sites and regions (Supplemental Information 4–6).

Longitude (correlation $= 0.867$, $p = 0.001$) was highly correlated with observed spatial patterns in contradiction to latitude (correlation $= 0.552$, $p = 0.021$). This result indicates that the relative position of each site along the Venezuelan coast (i.e., longitudinal variation), is an important factor to determine the features of coral assemblages in Venezuela, instead of the proximity to the coast (i.e., latitudinal variation).

## Patterns of beta diversity

When assessing $\beta$-diversity, we found the highest variation in species presence/absence occurring between transect of the same site ($F = 2.764$, $p = 0.045$) and between sites of the same locality ($F = 4.438$, $p = 0.026$). On the other hand, at larger scales, we found no significant dispersion in species composition between localities of the same region (Table 3, Supplemental Information 7–9). In addition, site and locality were the spatial scales with the largest Jaccard dissimilarity, with turn-over component as the main contributor. Furthermore, at larger scales, Jaccard dissimilarity decreased and the contribution of turn-over and nestedness became evener (Fig. 4). Thus, our result clearly shows that in Venezuela it is more likely to find changes in coral species composition at small to medium scales (i.e., hundreds of meters to tens of kilometers) than at larger scales (i.e., hundreds of kilometers). Finally, the results indicate that coral species found between the western, central and eastern region of Venezuela can result either from species replacement or from species loss, which is interpreted as a subset of a total pool of species.

## DISCUSSION

While coral assemblages have been extensively studied in Venezuela, this is the first multi-scale assessment to show the importance of spatial scales in determining the structure of these communities. Overall, we found that coral assemblages in Venezuela are variable from hundreds of meters and hundreds of kilometers. Additionally, the largest changes in the composition of coral species occurred at a small scale with a clear predominance of species turn-overs. Also, longitude and latitude are a good predictors of coral assemblage

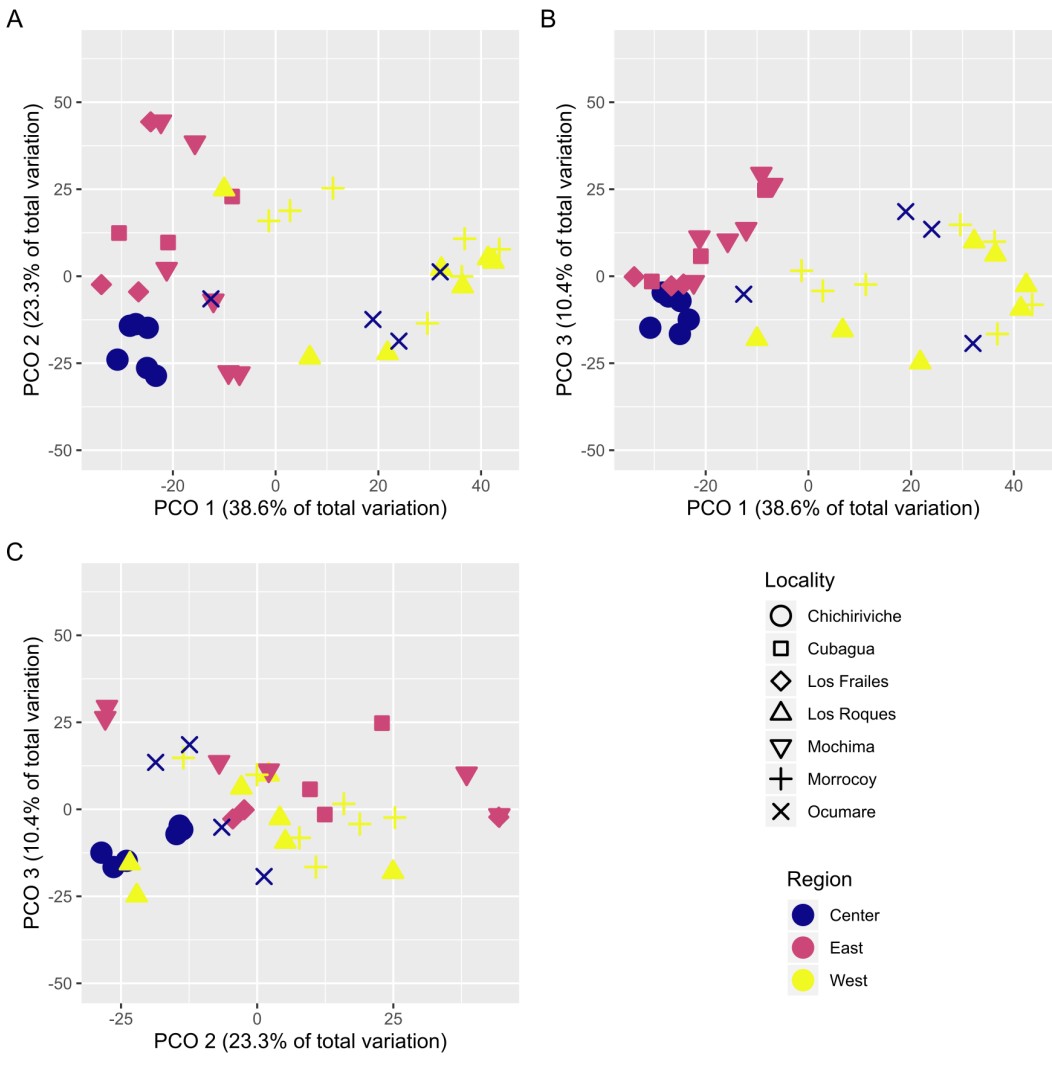

**Figure 2   Principal coordinates ordination plot (PCO) showing the variation of coral assemblages at the site level nested into locality.** The distance between samples are interpreted as % of similarity. (A, B, and C) represent the different combined pairs of the first three principal coordinates axes.

structure (i.e., species composition and abundance) further indicating that large-scale processes are also important to determine the structure of these communities.

## Patterns of coral assemblage structure

Previous studies have acknowledged the importance of spatial scales on coral assemblages in the Caribbean and in Venezuela (*Weil, 2003*). Particularly, the effect of upwelling and other related oceanographic processes has been pinpointed as strong factors that shape coral assemblage structure along the Venezuelan coast where at least 12 upwelling points have been targeted (*Miloslavich et al., 2003*; *Chollett et al., 2012*). Our study shows that coral assemblages in Venezuela are much variable within and between localities than we originally expected. We found two fold higher variability at small to medium scales when

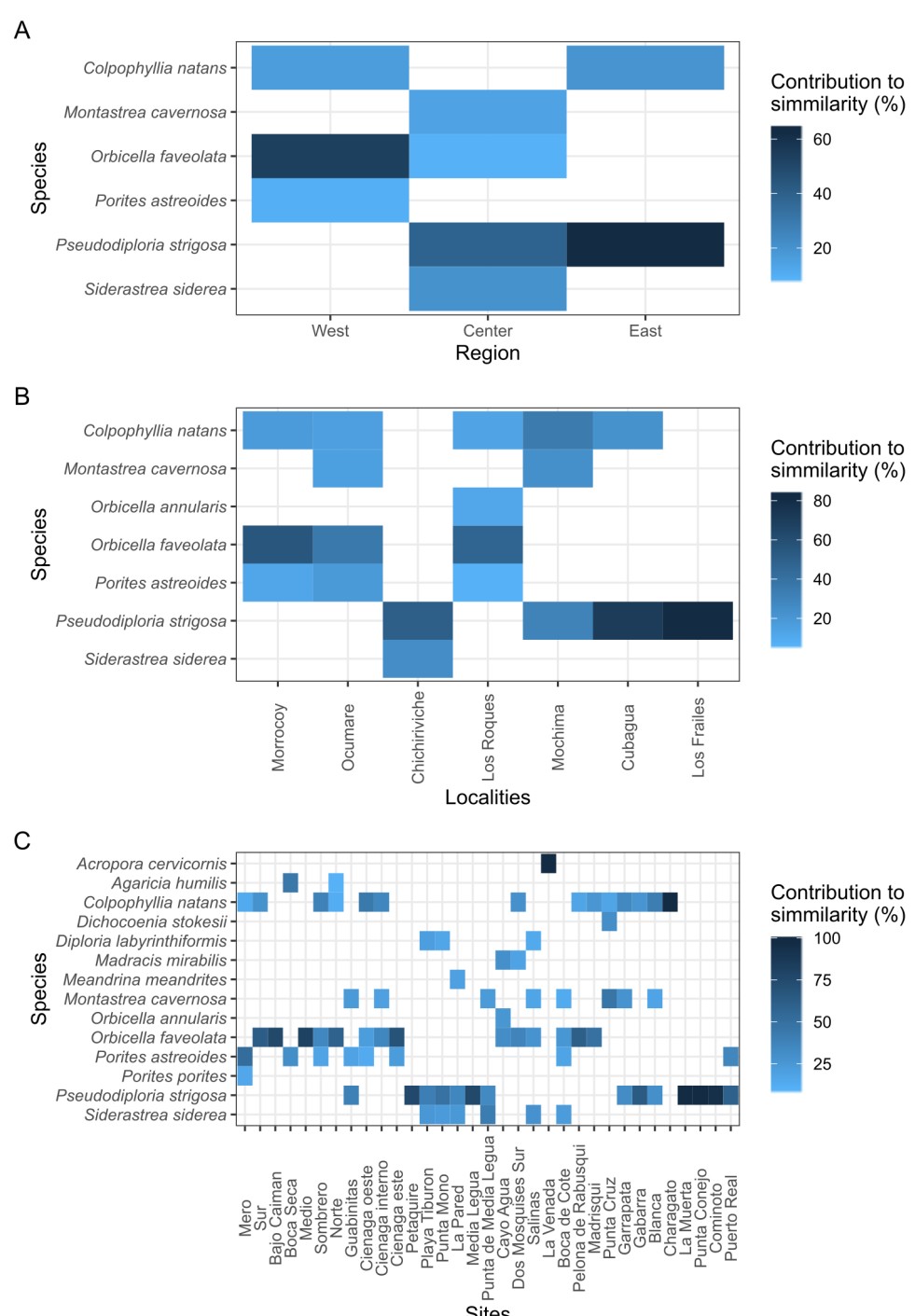

**Figure 3  Heat map showing the species contribution to similarity (Cut off for low contributions = 75%).** (A, B, and C) represent the three scales considered in the study.

**Table 3  Test of homogeneity of dispersions (PermDisp) to compare the distances from each factor to centroids as a test for similarity in $\beta$ diversity among factors.**

|           | Df  | F     | N.Perm | P(perm) |
|-----------|-----|-------|--------|---------|
| Site      | 35  | 2.764 | 999    | 0.045   |
|           | 107 |       |        |         |
| Locality  | 6   | 4.438 | 999    | 0.026   |
|           | 29  |       |        |         |
| Region    | 2   | 8.216 | 999    | 0.283   |
|           | 4   |       |        |         |

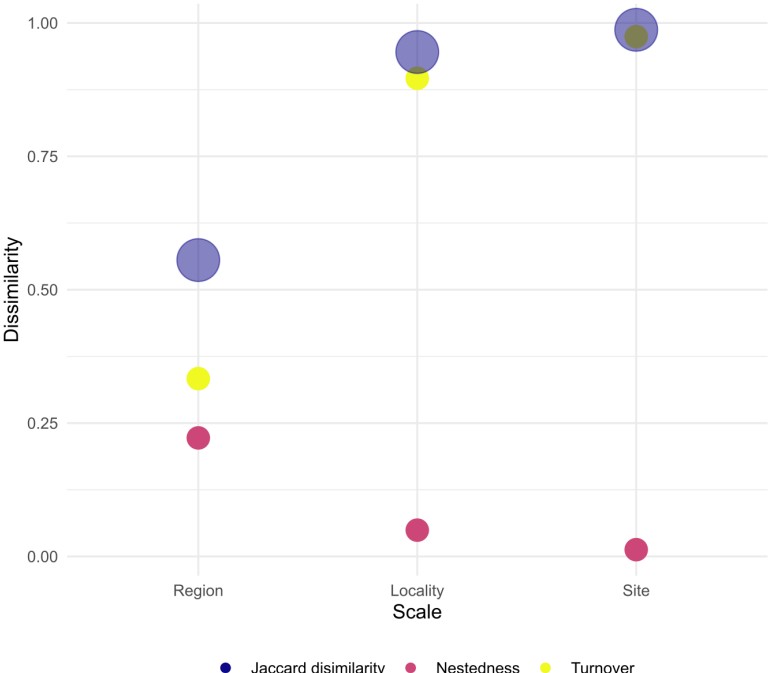

**Figure 4  Total dissimilarity as Jaccard index and their respective turn-over and nestedness components for each spatial scale.**

compared to regions. However, coral assemblages between the western, central and eastern regions differed by 21.35%, further indicating that differences at scale of region cannot be neglected. Moreover, about 32% of the total variance in coral species composition and abundance was associated to the residual which indicates that other variables like levels of anthropogenic disturbance, oceanic influence or other intermediate scales between those taken into consideration might also be relevant to determine the structure of these assemblages.

In Venezuela, encrusting communities associated with mangrove roots have been studied following a spatial hierarchical design (*Guerra-castro, Conde & Cruz-motta, 2016*). Similar to our results, higher variation for this assemblages were found at the smallest and biggest scales (*Guerra-castro, Conde & Cruz-motta, 2016*). Moreover, algal assemblages

associated rocky platforms have previously been found to be highly variable at tens of kilometers and not between localities or regions, further illustrating the importance of local processes in providing structure to different assemblages of sessile organisms in the country (*Cruz-Motta, 2007*).

Only a few numbers of studies encompassing multiple hierarchical spatial scales have been conducted in the Caribbean. For example, in a multi-scale study *Williams et al. (2015)* surveyed a series of reef sites across locations and different bio-regions in the Caribbean for three major coral taxonomic groups: corals, sponges and octocorals. They concluded these faunas exhibited considerable biogeographical variability at broad spatial scales (hundreds of kilometers). However similar to our study, they found a higher degree of variability within sites highlighting the relevance of local ecological drivers (e.g., rugosity and wave exposure) in structuring coral assemblages. Other studies have also taken into account the importance of spatial scales for coral assemblages, but must of them have focused on total live coral cover (*Murdoch & Aronson, 1999*) and total abundance of colonies (*Hughes et al., 1999*).

It is widely acknowledged that behind patterns are ecological processes that shape communities (*Barton et al., 2013*). In coral reef ecosystems, processes such as predation (e.g., herbivory) and competition have profound impacts on species abundance and composition at scales of a few meters. For example, very high densities or the absence of *Diadema antillarum* can determine the composition of corals in patches of few m$^2$ (*Sammarco, 1980*). Also, reef fish can preferentially prey on certain coral species, thus decreasing their abundance or making them less competitive than other neighboring corals. In addition, the presence of vermetids could potentially modify the survival rates of coral species (*Lenihan et al., 2011*). Furthermore, factors such as structural complexity (e.g., rugosity and micro scale habitat heterogeneity) may influence competition and survival of colonies depending on their sizes (*Almany, 2004*; *Zilberberg & Edmunds, 2001*). Coral reefs in Venezuela are known to be highly variable within and between sites but the processes responsible for these patterns have not been firmly established. However, spatial variation of coral assemblages in Los Roques has been associated to changes in reef slopes (*Cróquer & Villamizar, 1998*).

Various anthropogenic stressors can alter biological interactions thereby affecting the structure of coral assemblages. For example, overfishing often leads to the dominance of macroalgae which escape to herbivory control (*Hughes, 1994*). The selective extraction of species of carnivorous fish can lead to an increment in the abundance of echinoderms, which may also steer to increasing erosion, loss of topographic complexity and live coral cover (*Roberts, 1995*; *Mcclanahan & Muthiga, 1988*). Also, high intensity of recreational activities represents an important disturbance to marine communities (*Milazzo et al., 2002*), for example, coral cover and the proportion of massive corals is being found to be lower in places with high recreational diving intensity (*Tratalos & Austin, 2001*). Thus, spatial variability recorded within sites of Morrocoy and Los Roques National Parks could be explained by their differences in touristic use for not all sites within these MPAs are exposed to the same human pressures.

In addition, experimental evidence shows that some coral species differ in resistance to environmental stressors such as sedimentation (*McClanahan & Obura, 1997*; *Rogers, 1983*) which could explain the patterns observed within sites and between locations showing higher sedimentary regimes and river presence (e.g., coastal versus oceanic reef sites) (*Dikou & Van Woesik, 2006*). Furthermore, local oceanographic events can generate mortality which leads to changes in the structure of coral assemblages. Differences between Playa Caimán, Cayo Norte and Sombrero (Morrocoy National Park) represents a example of how abnormal oceanographic conditions can alter benthic communities by killing dominant species in specific sites while promoting stable alternative states which hampers recovery (*Bastidas, Croquer & Bone, 2006*). Our results seem to support that each site/location in Venezuela possess different communities because they may have been affected/unaffected by different stressors and/or mortality events in different times. Thus, high spatial variability on coral assemblages in Venezuela could be related to the differences in the disturbance regime and local history as noticed in other studies (*Pisapia, Burn & Pratchett, 2019*).

Coral bleaching mortality events may be patchy (*West & Salm, 2003*) and could potentially affect coral assemblages at different spatial scales (e.g., within sites, localities and regions). For example, in 2010 an increase in seawater temperature in Los Roques National Park affected 72% of the colonies at the study sites, showing bleaching and prevalence of diseases such as black band and white plague. Extensive mortality caused changes in the community structure a year later (*Bastidas et al., 2012*). Other bleaching events recorded in Venezuela since 1998 primarily affected reefs in several oceanic islands and the western and central coast of Venezuela, but these events did not produce extensive mortality events like the one reported in 2010 (*Rodríguez et al., 2010*; *Bastidas et al., 2012*; *Mónaco et al., 2012*).

It is likely that the eastern coast of Venezuela remained less affected by bleaching events because of seasonal upwelling and unique environmental conditions. In fact, *Nakamura & Van Woesik (2001)*] found differences in coral mortality rates during bleaching events, according to local environmental settings (e.g., light intensity, penetration, temperature and currents). However, *Chollett, Mumby & Cortés (2010)* argued that upwelling does not necessarily guarantee a refuge for corals. Thus, in Venezuela differences between geographical regions could be strongly influenced by factors such as nutrient input and temperature decrease associated with the upwelling season (*Birkeland, 1988*; *Weil, 2003*). In fact, it has been suggested that during these periods the assemblages of macroalgae become more dominant (*Diaz-Pulido & Garzón-Ferreira, 2002*) which could modify the coral assemblage structure through competitive processes.

The Venezuelan coast is characterized by an upwelling period that occurs between January and June, in particular, the eastern region of the country is characterized by a large area of upwelling (*Castellanos, Varela & Muller-Karger, 2002*). However, studies such as those of *Jiménez & Cortés (2003)* and *Rodríguez et al. (2009)* did not find an effect of the upwelling on factors such as the reproductive behavior of spawn corals and the growth rates of colonies. On the other hand, the dynamic of black band disease, one of the most important factors producing rapid coral mortality in Cubagua has been shown to be deeply influenced by upwelling events (*Rodriguez & Croquer, 2008*). Thus, our results indicate
that upwelling alone is not sufficient to explain the extremely variable nature of coral assemblage.

## Patterns of beta diversity

We found that differences in coral species composition occurred at spatial scales of hundreds of meters to tens of kilometers. Although it is known that $\beta$-diversity depends on the spatial scale at which it is measured, opposite to our results, in most studies $\beta$-diversity is assumed to be homogeneous at small spatial scales (*Hewitt et al., 2005*; *Whittaker, 1975*). Changes in species composition at scales of tens of meters often occurs in highly-heterogeneous habitats sampled with enough resolution to detect these changes (*Hewitt et al., 2005*). In our study, we found more likely to have different species composition within transects of a single site than between localities belonging to different regions. Thus, our results clearly support that coral habitats in Venezuela are extremely variable at local scales, suggesting significant environmental heterogeneity within reef habitats, with coral species probably forming mosaics or patches within a single habitat. However, it is not clear what are the conditions favoring this heterogeneity within the reef sites. This variation at the smallest scale could mean that Venezuelan coral assemblages are in good condition, as well as there are patches of mortality, with more or fewer species, reflected in a high turnover rate.

At larger scales (i.e., between the eastern, central and western Venezuelan coast) we found quite similar and homogeneous coral species composition, which may be partly due to the reduced pool of species that exist in the Caribbean when compared to the Indo Pacific region. In regions with larger species pools such as the Indo Pacific, $\beta$-diversity tends to be higher at larger spatial scales because species represent a subset of a total species pool, e.g., *Zvuloni, Van Woesik & Loya (2010)*. In Venezuela, coral reefs located at the oceanic sites and the western coast are dominated by *Orbicellas* , whereas in the eastern coast and the majority of sites located at the central coast *Pseudodiplorias* and *Colpophyllias* become more important.

Our results indicate that at scales of tens of kilometers species nestedness (loss) becomes as important as species turnover (replacement). These two components arise from different ecological phenomena (*Baselga, 2010*). Species nestedness occurs when the biotas of sites with smaller numbers of species are subsets of the biotas at richer sites, reflecting a non-random process of species loss as a consequence of any factor that promotes the orderly disaggregation of assemblages (*Baselga, 2010*). On the other hand, species turn-over (replacement) occurs as a consequence of environmental sorting and spatial and historical constraints (i.e., stochastic process) (*Qian, Ricklefs & White, 2004*). For example, processes such as settlement selectivity of coral larvae could explain species turn overs at tens of meters and kilometers. Coral larvae are known to select certain characteristics in the habitat to settle down, e.g., presence of certain species of coralline algae (*Ritson-Williams et al., 2010*) or sounds of the reef (*Vermeij et al., 2010*). Our results therefore indicate that coral assemblage structure in Venezuela is probably regulated by a series of interconnected processes acting alone and/or in combination at various spatial scales. This result highlights the importance of creating scale-adapted management actions in Venezuela since the smallest scales reflect the greatest variability. However, very small MPAs are often ineffective in achieving their

conservation goals, so they must necessarily be chained into a large-scale strategy (*Agardy, Di Sciara & Christie, 2011*).

## CONCLUSION

In summary, coral assemblage structure in Venezuela is highly variable at different spatial scales but within locality variability seem to be very important. The processes that could underlie these patterns are diverse and complex and little experimental efforts to untangle the specific contribution of each factor have been conducted. Longitude is a good predictor of coral assemblages in Venezuela. Upwelling-related processes could be targeted as potential candidates to explain longitudinal variation of coral assemblages, whereas oceanographic/coastal processes could explain latitudinal variability. Regarding $\beta$-diversity, coral assemblages are fairly homogeneous across the Venezuelan coast, while increasing spatial resolution shows greater heterogeneity, with smaller scales revealing a greater change in species composition. In addition, the replacement of species is a relevant phenomenon to explain these diversity patterns. Results from this study highlights the importance of taking into account local variability during the design and implementation of specific conservation efforts.

## ACKNOWLEDGEMENTS

We thank the support provided by Fundación para la Defensa de la Naturaleza (FUDENA), Instituto de Estudios Avanzados (IDEA) and the Escuela de Ciencias Aplicadas del Mar (ECAM), as well as José Cappelletto, Zlatka Rebolledo and Alejandra Verde who helped with the field work. We also thank Rita Peachey and all the 39th AMLC meeting committee for their help in attending the conference.

### Funding

This work was funded by Waitt Foundation through the Rapid Ocean Conservation grant (ROC). The funders had no role in study design, data collection and analysis, decision to publish, or preparation of the manuscript.

### Grant Disclosures

The following grant information was disclosed by the authors:
Waitt Foundation through the Rapid Ocean Conservation grant (ROC).

### Competing Interests

The authors declare there are no competing interests.

### Author Contributions

- Emy Miyazawa conceived and designed the experiments, performed the experiments, analyzed the data, prepared figures and/or tables, authored or reviewed drafts of the paper, and approved the final draft.

- Luis M. Montilla performed the experiments, prepared figures and/or tables, authored or reviewed drafts of the paper, and approved the final draft.
- Esteban Alejandro Agudo-Adriani and Gloria Mariño-Briceño performed the experiments, authored or reviewed drafts of the paper, and approved the final draft.
- Alfredo Ascanio performed the experiments, authored or reviewed drafts of the paper, wrote the scripts for data manipulation, and approved the final draft.
- Aldo Croquer conceived and designed the experiments, performed the experiments, authored or reviewed drafts of the paper, and approved the final draft.

## Field Study Permissions

The following information was supplied relating to field study approvals (i.e., approving body and any reference numbers):

The Ministerio del Poder Popular para el Ecosocialismo y Aguas. Dirección General de Diversidad Biológica granted permission to take pictures at marine protected areas (Oficio N° 0033).

## Data Availability

Raw data is available as a Supplementary File.

## Supplemental Information

Supplemental information for this article can be found online at http://dx.doi.org/10.7717/peerj.9082#supplemental-information.

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
