# Peer review of "On the importance of spatial scales on beta diversity of coral assemblages: a study from Venezuelan coral reefs"

_PeerJ, doi:10.7717/peerj.9082_

## Round 0.1 · original submission · Major Revisions

Three expert reviewers have evaluated your manuscript and their comments can be seen below and in an attached PDF. While there is agreement that this is an important manuscript, however, all reviewers have brought up some major issues that need to be clarified. Therefore, I encourage you to take into account these observations and in your resubmission, ensure that you respond to all of the comments made.

Reviewer 1 ·

Basic reporting

1. English used throughout the manuscript is clear and unambiguous.

2. Literature references are complete. They are sufficient and adequate to the topic presented and discussion of results.

3. Professional article structure, figures, tables. Raw data shared.

The authors comply with the structure that is required for this type of article, where they include relevant and complementary images to the manuscript, as well as raw data shared in a CSV text file.
However, the authors should consider next comments for some figure captions for clarified the information presented.
Figure 1: Mention which sites correspond to each region.
Figure 2. Mention that in all PCO ordinations, the variation of coral assemblages is shown at the site level nested into locations.
As a suggestion, I have seen in the whole manuscript, which is used "coral community" and "coral assemblage" in a similar way. I believe the most appropriate is to use "coral assemblage" instead of “coral community” for avoiding confusion.

4. Self-contained with relevant results to hypotheses.
The hypothesis of the article is presented in the lines 95-97. However, I consider this hypothesis very simple and does not provide enough details of coral beta diversity variation concerning the presence of different environment setting described along the Venezuelan coast. The hypothesis should be more explicit, where the particular hypotheses of each spatial scale considered in the experimental design are described.

Experimental design

This work conforms to the aims and scope of the Peer J journal.

This work provides relevant information to know the variation of the composition and coverage of coral species in the western Caribbean. However, their approach and experimental design are not adequate to answer the questions asked in this work.
I consider that this study has a rigorous investigation performed to a high technical and ethical standard, based on non-extractive benthic surveys following the guidelines by the Global Coral Reef Monitoring Network-Caribbean (GCRMN), and using the photoQuad software to identify coral species and their coverage.

Methods and data analysis were described without enough details, and it is not totally possible to replicate them.

Lines 117-123. The proposed experimental corresponds to a fully-nested and unbalanced multifactorial model, where the factors were: i) Region with three levels (West, Center, East); ii) Location with seven levels (2 to 3 localities per region); iii) Reef sites, 4 to 7 per locality; iv) Four transects per reef site. Sampling effort was obtained with 60 photos per site (15 per transect).
It is necessary to mention there are localities with coastal and oceanic influence, and some of them are continental or insular. The oceanic locality has different oceanographic conditions than coastal localities, as mentioned in the discussion section. Likewise, there is a difference in the environmental conditions between continental or insular localities. On the other hand, is the human impact different across all localities and reef sites? It is known that anthropogenic impact can affect the variation of coral assemblages. Therefore, it is necessary to explain, if the analysis of the fully-nested model of the spatial variation (i.e., longitude and latitude) of the coral assemblage, includes the natural variation of the sites when considering their coastal/oceanic, continental/insular influence, as well as the effect of human disturbance.

Inline 136, it is mentioned that two types of matrices were obtained using photo-quadrats, one with the relative abundance of coral species, and the other with data on the coral species presence/absence. For this, I reviewed the raw database that the authors included along with this article; thus, I have several comments:

i) Why were relative abundances of the species used and not the absolute abundance that represents the current live cover of the coral species? I believe that the absolute cover represents the current conditions of coral assemblage, as well as the change of its beta diversity across different spatial scales.
ii) When reviewing the raw database, I found that the lowest values per species are 20% relative abundance. I consider that this value is high for those species of low abundance and restricted distribution, which can be considered as rare. The latter species are those that most contribute significantly to beta diversity.
iii) The taxonomic resolution of the identification of any species is fundamental for the analyzes that evaluate the variation of its beta diversity. In the raw database there are the following categories: Other (Scleractinian encrusting), Other (Scleractinian foliose), Other (Scleractinian massive), Other (Scleractinian plate), and Other (Scleractinian solitary / other); as well as several genera of coral with undetermined species, which can be considered as coral taxa. In the case of the "Other" categories, each category is considered as a particular taxon in the analyzes; however, it is not known how many taxa exist within these categories. Although in these categories, the relative abundances are low, and the frequency is few, numerically they are relevant when analyzing beta diversity because this potentially increases the species turnover. In this work, how is it justified to use the database with these categories, and how can this affect the estimation of the variation of the coral species beta diversity?
iv) Photo-quadrats were used as true replicas nested within the transects. However, about 50% of 2,134 photo-quadrats have zero relative abundances, since I consider that no species or taxa of corals were recorded. I believe that this is a serious problem since there are a large number of zeros in the replicas with which the average dissimilarities of the species in the multi-scale nested design are estimated, which should affect the estimation of beta diversity. One option to reduce this bias is to use these photo-quadrats as Pseudo-replicas nested within the transects. Thus, the transects could be used as true replicas with the composition and average absolute coverage values of coral species. If this option is considered, a spatial scale would have to be eliminated in the experimental design. Thus, it would only be:
Y = Region + Localities(Region) + Sites(Localities (Region)) + error
If the authors consider not to follow this suggestion, then they should justify how this large number of zeros in the samples (photo-quadrats) does not numerically affect the estimation of the dissimilarities of the species across the different spatial scales analyzed.

Lines 139-145. It is necessary to describe the pre-treatment of data to be able to build the similarity matrices of Bray-Curtis to develop the PERMANOVA model and PCO ordination.
I want to comment that in order to carry out the analysis of the beta diversity assessment, it is essential to know which species favor the species turnover rate, or well, the species dissimilarities within each spatial scale studied. In this study, PCO ordinations were only used to visualize the change in species similarity at the reef site level, where Pearson correlations> 0.7 were used to identify which coral species contribute most to the structure of coral assemblage. I consider that to complement this analysis (PCO), it is necessary to perform a similarity percentage analysis (SIMPER) based on the same experimental design (fully-nested model), pre-treatment and similarity matrix used in the PERMANOVAs. This SIMPER is to identify and estimate the species that contribute most to the beta diversity (average dissimilarities) change at each spatial scale with significant differences (see PERMANOVA Table).

Lines 146-147. The information described on how the canonical analysis of principal coordinates (CAP) was done only mentions that it was done with the coverages of scleractinian corals and their latitudinal and longitudinal position. This information is not enough to understand how the CAP design was carried out since it is not known if the CAP was performed at the level of photo-quadrats, transects, reef sites, or locations. What kind of species similarity was used to perform the CAP?
I believe that the results of the PERMANOVA model should be used to make this analysis (CAP) and use the spatial scale that most contributed to the explanation of the variation in the change in the dissimilarity of coral species. In this case, the spatial scale of the reef sites can be used.
On the other hand, why were latitude and longitude only used as explanatory variables in the CAP? There are many variables associated with them, such as the upwellings along with the change in longitude, and the oceanographic variables associated with the latitude. Is it possible to obtain information on these variables (upwellings and oceanographic variables), perhaps with values on an ordinal scale, to make a more robust analysis? It would be interesting to know if the authors could have other environmental variables at the reef site scale, in order to improve data analysis, such as rugosity (topographic complexity), depth, other structural elements of the habitat (eg, live coral cover, macroalgae cover , sponges, turf and other types of substrates), and human impact variables. These kinds of variables can be used as predictive variables to explain the variation of the beta diversity of coral assemblage? With this information, it is possible to perform analyzes such as a canonical redundancy analysis (RDA) or a canonical correspondence analysis (CCA).

Lines 149-150. The differences in beta diversity were tested across different spatial scales using a test of homogeneity of dispersions (PermDisp) based only on presence/absence Jaccard similarity, based on the criteria of Anderson et al. (2006). But to complement the results of PERMANOVA, why was the Bray-Curtis similarity not used to carry out the PermDisp? In order to be able to infer whether the variation of the change in the assembly of coral species is given by location effects, dispersion effect or both.
On the other hand, when reviewing Table 2, the results of each PermDisp by spatial scale correspond to a one-way analysis. How can its use be justified, since the fully-nested design cannot be incorporated in the way the analysis was done? For example, I mean that by observing the significant variation of the smaller scales, and when analyzing the result obtained at the level of transects, PermDisp analyzed and tested the homogeneity of the dispersions of all transects but without considering the structure of the nesting: Transects(Sites (Localities (Regions))).
In the case of the analysis of beta diversity partition based on the strategy of Baselga and Orme (2012), it was performed in the same way as PermDisp, because the change in beta diversity, and their respective components, were only tested by spatial scale regardless of the nesting design. It would be interesting to know how the contributions of the species turnover and nestedness change within each region.
Finally, it is necessary to mention which were the samples used to develop the PermDisp and the partition of beta diversity by spatial scale. For example, were transects used to analyze the spatial scale of reef sites? Were the sites used to analyze the locality spatial scale? If so, the nested design was used in some way.

Validity of the findings

The results are well written and match what was found in the data analysis. However, if the authors decide to make some of the suggested changes in the used methods, some of the results of the different numerical analyzes may change.

Lines 178-189. The results of the PermDisp and beta diversity partition show that small and local scales are the most important to explain the variation of beta diversity, which suggests that environmental and spatial heterogeneity at this scale are essential to explain the coral species turnover. In that sense, it is important to consider other data analyses that assess the relationship of the change in the composition and cover of coral species with environmental variables (habitat structure, human or other) on a local scale (e.g., reef sites). Likewise, it is relevant to know which species favor dissimilarities (or beta diversity) and with which environmental variables they are related.
The discussion section is well written, and a good relationship of the results is made with published literature. However, this section may change if the reviewers consider making some changes to the methods used.

Line 206-209. What are the other spatial scales associated with residuals?

Lines 284-285. There is no analysis in this work to be able to accept the statement that upwelling alone is not sufficient to explain the extremely variable nature of the coral assemblage.

Lines 292-295. I suggest giving greater emphasis to the importance that coral assemblages have greater beta diversity on local scales. This suggests that there may be significant environmental heterogeneity within reef habitats, which could mean that coral assemblages still have adequate conditions on the coasts of Venezuela in comparison with other Caribbean areas.

Line 300. Delete the parenthesis in the year of the appointment (Zvuloni et al. 2010).

Line 320-322. It is mentioned that "upwelling-related processes could be targeted as potential candidates to explain longitudinal variation of coral communities, whereas oceanographic/coastal processes could explain latitudinal variability". While on lines 284-285, the opposite is mentioned: "upwelling alone is not sufficient to explain the extremely variable nature of coral community". What is the correct position of the authors concerning the relationship of the upwelling effect on the coral beta diversity? I recommend exploring whether it is possible to evaluate the effect of these variables (upwelling effect and oceanographic/coastal processes) on the variation of coral assemblage.

Additional comments

The work performed by Miyazawa et al., entitled “On the importance of spatial scales on beta diversity of coral communities: a case study from Venezuelan coral reefs”, makes a multiscale analysis to identify the greatest variability of the coral assemblage beta diversity on the coast of Venezuela. For this, the authors carried out fieldwork where they obtained a considerably extensive sampling effort in different regions, locations, and sampling sites.

Their results are interesting and make a relevant contribution to scientific knowledge and the understanding of the processes of the variation of species turnover in coral ecosystems. However, I found some troubles in the section of materials and methods. I consider that this article should clarify some details of the data analysis and the results, as well as other details that should be addressed by the authors. Therefore, I consider that this article cannot be published in the form as it currently is. Authors could consider doing several corrections I have suggested.

Reviewer 2 ·

Basic reporting

no comment

Experimental design

no comment

Validity of the findings

no comment

Annotated reviews are not available for download in order to protect the identity of reviewers who chose to remain anonymous.

Reviewer 3 ·

Basic reporting

several issues, see below

Experimental design

several issues, see below

Validity of the findings

several issues, see below

Additional comments

The subject of the ms is important for many aspects related to biodiversity, however this ms has some apparently serious issues that should be resolved

Estimating β-diversity requires an adequate estimation of the α-diversity of coral assemblages to be compared. However, at local scales α-diversity (transect? site? in this study) may differ due to location, reef, reef zone, zone patchiness, community history, environmental setting, biological interactions and catastrophic events; as elsewhere in the reef ecosystem.

Since adequate estimates of species diversity are directly related to sampling effort, the sampling design should take into account the potential sources of variability at the different sampling sites, and its effect on sufficient sample size. One way to do this is to find a sample size that will be sufficient in all instances, thus allowing for direct sample comparison.

However, this study is based on a standard sampling protocol and the reader wonder if the sampling was adequate to properly address the questions posed. I believe that addressing in detail the following points would give credibility to the study:
1. How similar were the seven study sites in terms of reef zonation, exposure and general setting?.
2. Why CHI site is not considered as part of the East region?
3. Demonstrate that transect length was adequate for each surveyed site and that four transects is a sufficient number of such transects per site
4. Please explain the spatial layout of the transects on each site and how such layout was decided
5. Please explain the pros and cons of a systematic transect sampling, instead of a random one to assess diversity.
6. Please explain why 15 photos were taken per transect instead of the 30+ that could be obtained (NOTE: regarding the assertion on line 128 that transect photos of the same transect are replicates: As images are taken along a limited space along the transect, instead of randomly on the study site, they are pseudo-replicates; true replicates would implicate a random layout).

An issue of great concern is related to the above in conjunction with a unit sample size image of 80 by 90 cm, image that is further subsampled by 25 random points. Perhaps if communities were composed by rather small size colonies such approach would be OK, but not so if coral assemblages are dominated by relatively large colonies because these will occupy large image sections or whole images and this will further reduce sample size for diversity estimation. This may be the case in this study as the ms indicates that Orbicellas, Pseudodiplorias and Colpophyllias dominate (line 302-303, Fig 2).

As this is a rather important concern I tried to resolve this doubt by means of the supplemental raw-data sheet provided, however the *.csv file has a sort of continuous line format useless to discriminate species/sample info (so, it is not truly supplementary materiel). Nevertheless, is possible to observe a large quantity of zeros in every line, presumably corresponding to different species values, further increasing the above concern. On the same tenet relative homogneization at the regional levels could also be a result of sufficient sample size. All these require a detailed clarification by authors.

Given the above, I consider that the above questions have to be properly addressed to consider publishing this study

However, a few more comments to the authors:
• An issue of over dispersion may severely affect the permanova analysis, so it is necessary to validate such analysis.
• it may be that the transect factor level is difficult to justify. If so, I suggest to reanalyze with site as the lower level
• Latitudinal relevance could be related to island vs continental, perhaps a post hoc site comparison could help in clarifying this
• Justify the inclusion of correlation vectors on the PCO plots and its interpretation in that particular type of diagram
• Explain the values in the axes of the PCO diagrams and what each plot means
• It will be adequate to explain how a dispersion analysis could be applied to absence/presence data

While the ms is well written much of it has been said before. In the introduction by Beselga and several other authors; in coral community and β-diversity patterns sections by many other authors as well. I will suggest, once the methodological issues are resolved to concentrate in the homogenization of diversity problem and in this case to what extent estimates of nestedness and turnover could be a sampling artifact over valid estimates.

---

## Round 0.2 · Minor Revisions

Please take note of some final adjustments to the manuscript that need to be made.

Reviewer 1 ·

Basic reporting

The English used in this second version is from my point of view is clear and unambiguous.
This manuscript presents an adequate introduction that introduces to the main problem addressed by this work. All literature cited is necessary, and it was correctly cited.
The structure of the article (manuscript, figures, tables, raw data, and supplementary material) complies with what is requested in the instructions for the authors.
This work presents relevant results related to the hypothesis presented.

Experimental design

This second version used an experimental design that tests the proposed hypothesis and achieves the objectives of the work. This new experimental design is more explicitly understood why it is important to analyze the spatial variation of coral assemblage among different scales.
Statistical analysis models agree to this experimental design. The numerical methods used are correct to analyze the spatial variation of the coral assembly at multiple spatial scales.
The assessment strategy of coral beta diversity was also adequate to get important results on how species turnover and nestedness vary at different spatial scales.
This new version provides more detailed and clear information on the methods and analyses used, which allows us to replicate them.

Validity of the findings

The results of this work show that coral assemblages in Venezuela have an important spatial variation. Thus, their species composition and coverage were mainly explained by local and intermediate scales due to greater spatial heterogeneity. This result supports an important contribution of coral species turnover at these spatial scales. All the presented results are supported by several statistical analysis.
The conclusions are appropriately stated and based on the results and the stated hypothesis.
I found only a few minor details:
Line 82. Change the lowercase letter g of the word Gonzalez to a capital G.
Line 200. Remove the first parenthesis in ((Figure 3).
In Figure 3 use italic letters for scientific names of coral species.

Additional comments

First of all, I congratulate the authors of this article as they improved it considerably because they followed most of the suggestions made in the first version.
I consider that this version of the manuscript has the necessary quality to be accepted and published in Peer J, once minor corrections are corrected.

Reviewer 2 ·

Basic reporting

no comment

Experimental design

no comment

Validity of the findings

no comment

Additional comments

no comment

---

## Round 0.3 · accepted · Accept

I am satisfied with the changes made to the manuscript.